# Cell-Specific Transport and Thyroid Hormone Receptor Isoform Selectivity Account for Hepatocyte-Targeted Thyromimetic Action of MGL-3196

**DOI:** 10.3390/ijms232213714

**Published:** 2022-11-08

**Authors:** Georg Sebastian Hönes, Ramona Gowry Sivakumar, Christoph Hoppe, Jörg König, Dagmar Führer, Lars Christian Moeller

**Affiliations:** 1Department of Endocrinology, Diabetes and Metabolism, Division of Laboratory Research, University Hospital Essen, University of Duisburg-Essen, Hufelandstr. 55, 45147 Essen, Germany; 2Institute of Experimental and Clinical Pharmacology and Toxicology, Friedrich-Alexander-Universität Erlangen-Nürnberg, Fahrstr. 17, 91054 Erlangen, Germany

**Keywords:** MGL-3196, Resmetirom, thyromimetic, thyroid hormone action, thyroid hormone analog, thyroid hormone transport, OATP1B1

## Abstract

Thyroid hormones (THs) and TH receptor-beta (TRβ) reduce hepatic triglycerides, indicating a therapeutic potential for TH analogs in liver steatosis. To avoid adverse extrahepatic, especially TRα-mediated effects such as tachycardia and bone loss, TH analogs with combined TRβ and hepatocyte specificity are desired. MGL-3196 is a new TH analog that supposedly meets these criteria. Here, we characterize the thyromimetic potential of MGL-3196 in cell-based assays and address its cellular uptake requirements. We studied the contribution of liver-specific organic anion transporters (OATP)1B1 and 1B3 to MGL-3196 action. The TR isoform-specific efficacy of MGL-3196 compared with 3,5,3′-triiodothyronine (T_3_) was determined with luciferase assays and gene expression analysis in OATP1B1 and OATP1B3 and TRα- or TRβ-expressing cells and in primary murine hepatocytes (PMHs) from wild-type and TRβ knockout mice. We measured the oxygen consumption rate to compare the effects of MGL-3196 and T_3_ on mitochondrial respiration. We identified OATP1B1 as the primary transporter for MGL-3196. MGL-3196 had a high efficacy (90% that of T_3_) in activating TRβ, while the activation of TRα was only 25%. The treatment of PMHs with T_3_ and MGL-3196 at EC_50_ resulted in a similar induction of *Dio1* and repression of *Serpina7*. In HEK293 cells stably expressing OATP1B1, MGL-3196 had comparable effects on mitochondrial respiration as T_3_. These data indicate that MGL-3196’s hepatic thyromimetic action, the basis for its therapeutic use, results from a combination of hepatocyte-specific transport by OATP1B1 and the selective activation of TRβ over TRα.

## 1. Introduction

Thyroid hormone (TH; T_4_, 3,5,3′,5′-tetraiodothyroxine, thyroxine, and its active form T_3_, 3,5,3′-triiodothyronine, thyronine) is essential for development and physiology, including hepatic triglyceride metabolism [1,2]. The uptake of T_4_ and T_3_ into target cells is mediated by plasma membrane transporter proteins. TH transporters are members of the monocarboxylate transporter family (e.g., MCT8 and MCT10), organic anion transporting polypeptides (OATPs), e.g., OATP1C1, and L-type amino acid transporters (e.g., LAT1 and LAT2) [3]. T_3_ acts by binding to the TH receptors (TRs) α and β that are expressed in virtually all cells. The TR isoform expression differs between cell types [4,5]. In hepatocytes, TRβ is the predominant TR isoform, while TRα is mainly expressed in cardiomyocytes and osteoblasts [4,6]. Therefore, local, cell-type-specific TH action is determined by TH transport into these cells and their TR isoform expression.

In the liver, local TRβ action lowers lipid content, thus offering a therapeutic potential of TH action in the treatment of non-alcoholic fatty liver disease (NAFLD) and non-alcoholic steatohepatitis (NASH). However, the systemic supra-physiological doses of TH lead to thyrotoxicosis with adverse effects on the cardiovascular system and the bone, mainly mediated by the activation of TRα. Therefore, TRβ agonists have been developed, e.g., GC-1 (Sobetirome), GC-24, KB-141, and KB-2115 (Eprotirome). Eprotirome was highly successful in reducing LDL cholesterol and triglycerides in patients with familial hypercholesterolemia in a phase 3 clinical trial. However, a significant increase in transaminases in patients and deleterious effects on the cartilage in dogs led to the termination of the clinical trial. In a phase 1 trial, Sobetirome treatment resulted in a reduction in LDL cholesterol by up to 41% but was terminated due to concerns over a class effect in light of the adverse effects of Eprotirome [7,8]. For these reasons, to date, no TRβ agonist obtained NIH approval despite its demonstrated beneficial effects on serum and hepatic lipid concentration [9,10]. The recent development of TH analogs aimed at better focusing TH action by combining hepatocyte specificity and TRβ selectivity, which would preserve the beneficial hepatic effects while reducing the extrahepatic adverse effects. 

Presumably, MGL-3196 (also called VIA-3196 or Resmetirom) meets these criteria [11]. MGL-3196 is a pyridazinone-based TH analog with an azauracil group containing a cyano substitution that increases TRβ selectivity (Figure 1). In preclinical studies, the oral application of MGL-3196 reduced serum cholesterol without affecting bone mineral density in mice or cardiac α-MHC expression in thyroidectomized rats [11]. Recently, results from a phase 2 trial of MGL-3196 in NASH patients and from a 36-week active treatment open-label extension study have been reported [12,13]. Relative to the baseline, 74 patients treated with MGL-3196 for 36 weeks in the main study showed a 37.3% reduction in liver fat compared with 8.9% of the 34 placebo-treated patients, determined by MRI, and reduced liver fibrosis, as determined by liver biopsy. This was accompanied by a significant reduction in serum LDL cholesterol and triglyceride concentration. No adverse effects such as the suppression of thyroid stimulating hormone (TSH) concentration, reduction in bone mineral density, or increased heart rate were reported. These promising results from the phase 2 study led to a phase 3 study (MAESTRO-NASH; NCT03900429).

MGL-3196 appears to have a favorable overall pharmacological profile with significant lipid-lowering efficacy in the absence of adverse effects. Presumably, this is due to a combination of TRβ and hepatocyte specificity, reducing extrahepatic, especially TRα-mediated, adverse effects. 

In a cell-free co-factor binding assay, MGL-3196 selectivity for TRβ was determined to be 28-fold with an EC_50_ of 0.21 µM compared with 3.74 µM for TRα [11]. Another study found a 12.5-fold selectivity for TRβ over TRα using HEK293T cells overexpressing retinoic X receptor-α (RXRα) in combination with either TRα or TRβ [14]. This study also revealed high variations in EC_50_ values between different cell lines, possibly explained by differences in MGL-3196 cellular uptake. The transport of TH and TH derivatives is facilitated by OATPs. Interestingly, the expression of OATP1B1 and OATP1B3 is restricted to the liver [3,15,16]. It was, therefore, hypothesized that MGL-3196 could be a substrate for OATP1B1 and OATP1B3, which would explain MGL-3196’s hepatocyte-specific action [14]. However, the requirement of OATP1B1 and/or OATP1B3 for the transport of MGL-3196 has not been studied to date. Studies on MGL-3196 action were mostly carried out either in cell-free systems or in HEK293 cells that do not express OATP1B1 or OATP1B3. We aimed to study the contribution of hepatocyte-specific OATP1B1 and OATP1B3 to MGL-3196 action and its TR isoform selectivity in a human cell model and primary murine hepatocytes.

## 2. Results and Discussion

### 2.1. Expression of TH Transporters and TRs in HEK293 Cells

To determine the role of hepatocyte-specific transporters in MGL-3196 action, we studied wild-type HEK293 cells in comparison to HEK293 cells stably expressing the hepatocyte-specific transporters OATP1B1 (HEK-1B1) and OATP1B3 (HEK-1B3). First, we measured the endogenous expression of OATP1B1, OTAP1B3, and TH transporters MCT8, LAT1, and LAT2 [17,18,19]. MCT8, LAT1, and LAT2 were all expressed in all three cell lines. OATP1B1 and B3 were not expressed in the wild-type HEK cells and were detected only in the respective stably transfected cell line (Figure 2A). 

Next, we determined the expression of TRα and TRβ in HEK293 cells and found a threefold higher expression of TRα compared with TRβ (Figure 2B). This TRα:TRβ ratio is comparable to that found in human and mouse kidneys [4,5]. To overcome this difference, we transiently transfected the three cell lines with plasmids encoding for TRα and TRβ. This significantly increased the expression levels of both receptors and equalized their expression (Figure 2B). 

### 2.2. MGL-3196 Transport Is Mediated by OATP1B1 

We used a DR4 luciferase reporter assay to assess MGL-3196 action in HEK-1B1 and HEK-1B3 cells either overexpressing TRα or TRβ in comparison to T_3_ [20]. For T_3_, we used concentrations ranging from 0.1 to 1000 nM and MGL-3196 concentrations from 10 to 10,000 nM. Luciferase activity was increased by T_3_ in all cell lines. While there was no difference in the T_3_-mediated increase in the luciferase action of OATP1B1- or OATP1B3-expressing cells, treatment with MGL-3196 resulted in a stronger increase in luciferase activity in HEK-1B1 cells than in HEK-1B3 cells (Figure 3A), indicating that OATP1B1 is the preferred transporter for MGL-3196. 

To confirm the relevance of OATP1B1 for MGL-3196 transport, we used the OATP1B1 inhibitor glycyrrhizinic acid (GA) [21,22]. Pretreatment with 100 µM GA had no effect on the luciferase signal in T_3_-treated wild-type HEK cells that did not express OAPT1B1, demonstrating that GA does not inhibit T_3_ uptake in the absence of OATP1B that is mediated by MCT8, LAT1, and LAT2. However, in HEK-1B1 cells, the T_3_ effect was significantly reduced by GA pretreatment (Figure 3B), demonstrating that OATP1B1 contributes to T_3_ transport, which is in accordance with previous reports [3,16,23]. Importantly, MGL-3196 induced a luciferase signal only in HEK-1B1 cells but not in wild-type cells. This induction was completely abolished by GA pretreatment (Figure 3B).

From these data, we conclude that MGL-3196 is predominantly transported by OATP1B1, while OATP1B3 and other TH transporters (e.g., MCT8, LAT1, and LAT2) contribute little at the studied doses. OATP1B1 differs from other members of the OATP family in its exclusive expression in hepatocytes. Therefore, the requirement of OATP1B1 for the cellular action of MGL-3196 possibly explains the high liver specificity of MGL-3196 [24]. These results cannot exclude a possible transport of MGL-3196 by other transporters, e.g., OATP1C1 and SLC17A4, into other cells outside the liver. However, no extrahepatic adverse effects of MGL-3196, e.g., on TSH serum concentrations in probands or patients [11,25] or the heart and kidney weight in mice have been reported so far [26].

### 2.3. MGL-3196 Predominantly Acts via TRβ

The activation of TRα and TRβ by T_3_ resulted in almost superimposable dose–response curves for both receptors, demonstrating that T_3_ is not a selective TR agonist, whereas MGL-3196 induced a stronger response with TRβ (Figure 3A). Compared with T_3_, the activation of TRβ reached 90% in HEK-1B1 and 60% in HEK-1B3 cells, respectively, compared with 25% and 20% for TRα (Figure 3A). Based on the T_3_-mediated receptor activation, we calculated the EC_50_ values of MGL-3196 in OATP1B1- and OATP1B3-expressing cells [27]. In HEK-1B1 cells, the EC_50_ of MGL-3196 for TRβ was 0.601 µM, compared with 0.007 µM for T_3_ (Table 1), which illustrates that in those cells expressing OATP1B1, MGL-3196 is about 100-fold less potent than T_3_. Interestingly, in OATP1B3-expressing cells, the EC_50_ value of MGL-3196 for TRβ was 17,660 µM, which is about 30 times higher than in OATP1B1-expressing cells, further demonstrating that OATP1B1 and not OATB1B3 is the predominant transporter for MGL-3196. Remarkably, in this cell-based luciferase reporter assay, MGL-3196 failed to reach the 50% activation threshold in TRα-overexpressing cells, precluding the calculation of EC_50_ values for MGL-3196 and TRα, which further demonstrates the much higher affinity of MGL-3196 to TRβ over TRα.

The EC_50_ value of MGL-3196 in HEK-1B1 cells was about three to nine times higher than previously reported in two independent studies (601 nM vs. 210 nM [11] and 73.1 nM [14]). In these studies, a cell-free FRET assay was used to determine the EC_50_ values. The probable explanation for this difference is the additional requirement of the cellular import of MGL-3196 in our cell-based luciferase reporter assay, which more closely resembles the physiological situation. Luong et al. also determined the EC_50_ values for MGL-3196 with luciferase assays in wild-type HEK293T cells transiently expressing TRβ and found a fourfold higher EC_50_ (2365.8 nM) than that found in our experiments [14]. This difference reflects the absence of the transporter OATP1B1 and demonstrates that transport needs to be considered for the action of TH analogs such as MGL-3196. Furthermore, in Huh-7 cells (established from a human well-differentiated hepatocyte-derived carcinoma cell line), the EC_50_ value was much lower at 303.1 ± 50.9 nM and closer to the EC_50_ values determined in HEK-1B1 cells. Interestingly, Huh-7 cells express OATP1B1 [28], which again underscores the importance of transport for MGL-3196 action [11,14]. 

### 2.4. Equivalent Induction of TH Target Genes by MGL-3196 and T_3_

Next, we compared the MGL-3196- and T_3_-mediated induction of endogenous TH target genes in HEK-1B1 cells overexpressing either TRβ or TRα. The cells were treated with the EC_50_ values of T_3_ (10 nM) and MGL-3196 (600 nM) for 24 h. The expression of krüppel-like factor 9 (*KLF9*) and phosphoenolpyruvate carboxykinase 1 (*PCK1*) was similarly increased by T_3_ in TRβ- and TRα-overexpressing cells (Figure 4A). MGL-3196 treatment led to the significantly stronger induction of *KLF9* and *PCK1* in the cells overexpressing TRβ than in those overexpressing TRα, which demonstrates its TRβ selectivity [11].

To verify these results in a more physiological model, we used the primary murine hepatocytes (PMHs) from wild-type C57BL6/J mice and TRβ^−/−^ mice from the same genetic background. The thyromimetic action of MGL-3196 was compared with that of T_3_ in PMHs treated for 24 h with the EC_50_ of T_3_ and MGL-3196. T_3_ and MGL-3196 led to an equivalent induction of *Dio1* and *Pck1* as well as the repression of the negatively regulated TH target gene *Serpina7* (Figure 4B). Neither T_3_ nor MGL-3196 led to a transcriptional regulation in PMH of TRβ^−/−^ mice. 

Notably, human cells express two OAPT1B isoforms (1B1 and 1B3), whereas in mice, only one isoform exists. The common murine orthologue of OATP1B1 and OATP1B3 is Oatp1b2, encoded by *Slco1b2* [24]. Interestingly, the expression of *Slco1b2* was induced by T_3_ and by MGL-3196 in a TRβ-dependent manner (Figure 4B). Moreover, in PMHs from TRβ^−/−^ cells, the expression of *Slco1b2* was significantly decreased, independent of the treatment regime, suggesting a strong TRβ-mediated influence on *Slco1b2* expression. Therefore, testing for residual TRα-mediated off-target effects of MGL-3196 in TRβ^−/−^ cells is not possible, as a lack of Oatp1b2 expression prevents proper MGL-3196 uptake. However, the induction of Slco1b2 expression by T_3_ and MGL-3196 suggests the possibility of a positive feedback loop of TH and MGL-3196 uptake into hepatocytes, which would further increase MGL-3196 accumulation in hepatocytes and, consequently, enhance its hepatocyte specificity. 

### 2.5. MGL-3196 Improves Mitochondrial Function

T_3_ increases mitochondrial function in HepG2 cells [29]. As HepG2 cells do not express OATP1B1 [28], we used the HEK-1B1 cell line to study whether MGL-3196 activates mitochondrial function as T_3_. HEK-1B1 cells were treated with a receptor-saturating dose of 100 nM T_3_ and 6 µM MGL-3196 for 48 h, and the oxygen consumption rate (OCR) was measured (Figure 5A). Both T_3_ and MGL-3196 increased basal respiration, maximal respiration, ATP production, and proton leak (Figure 5B–E). The increased basal respiration accounts for a higher energy demand of the cells that is accompanied by an increase in ATP production. The increased maximum rate of respiration in T_3_- and MGL-3196-treated cells reflects a higher oxidation rate of substrates and thus an increased metabolic rate. The increased proton leak could indicate mitochondrial damage. The spare respiration capacity and non-mitochondrial oxygen consumption rate were not significantly increased (Figure 5F,G). This analysis of oxygen consumption rate and respiration demonstrates that MGL-3196 functionally mimics T_3_. 

## 3. Conclusions

THs harbor therapeutic potential for liver diseases such as NAFLD or NASH. Yet, as THs act in virtually all organs and cells, the challenge is to harness the desired beneficial TRβ-mediated effects on liver lipid metabolism while avoiding adverse effects on the cardiovascular system and bone turnover, mostly mediated by TRα [10,30,31]. This could be achieved with a hepatocyte-specific TRβ agonist. The present data show that MGL-3196 action requires the expression of OAT1B1, a hepatocyte-specific OATP. Furthermore, the activation of TRβ by MGL-3196 reached 90% of that of T_3_ compared with only 25% with TRα, demonstrating MGL-3196’s TR isoform-specific efficacy. MGL-3196 elicits the same effects as T_3_ on gene expression and cell metabolism, stimulating respiration and ATP production. Therefore, the present results demonstrate that the local, hepatic thyromimetic action of MGL-3196 is a result of hepatocyte-specific transport and high TRβ specificity. These features serve to explain the efficacy of reducing hepatic lipid content in patients and the absence of extrahepatic adverse results in clinical studies. MGL-3196 is an example of how local control of TH action, here achieved through transport and receptor specificity, allows for the therapeutic use of TH action. 

## 4. Material and Methods

### 4.1. Chemicals

T_3_ (3,3′,5-triiodo-l-thyronine sodium salt; Merk, Darmstadt, Germany) and MGL-3196 (Cayman Chemical, Ann Arbor, MI, USA) were both dissolved in dimethyl sulfoxide (DMSO; Sigma Aldrich, Darmstadt, Germany) at a concentration of 10 mM. Glycyrrhizinic acid (GA; Sigma Aldrich, Darmstadt, Germany) was dissolved in DMSO at a concentration of 50 mM.

### 4.2. Cell Culture and Transfection

Human embryonic kidney cells (HEK293) stably overexpressing human OATP1B1 (HEK-1B1) and OATP1B3 (HEK-1B3) were contributed by Dr. Jörg König, and their generation has been described previously [32]. The cells were maintained in a DMEM containing 4.5 g/L glucose, 1% pyruvate (DMEM+ GlutaMAX, Gibco™, Thermo Fisher Scientific, Schwerte, Germany), 1% ZellShield (Minerva Biolabs GmbH, Berlin, Germany), and 10% FCS (Fetal Calf Serum, Gibco™, Thermo Fisher Scientific, Schwerte, Germany) supplemented with 800 µg/mL G418 (Capricorn Scientific, Ebsdorfergrund, Germany) [33]. Only those cells in a passage number below 7 were used for the experiments. Transient transfection with expression vectors for TRα and TRβ was performed as previously described [20], with few modifications. Briefly, polyethyleneimine (Pei, Polyscience, Hirschberg an der Bergstraße, Germany) was used as a transfection reagent in a Pei/DNA ratio of 3:1 using a total amount of 0.5 µg DNA/24-well and 2 µg DNA/6-well.

### 4.3. Luciferase Assay

HEK293 cells were grown to 60–80% confluence on a 24-well plate. The transient transfections of DR4-Luc reporter plasmid, RL-TK control plasmid, and plasmids encoding for TRα, TRβ, or empty vector pcDNA3 were carried out as previously described [20]. After transfection, cells were maintained under hypothyroid conditions for another 24 h using 5% TH-depleted FCS. TH-depleted FCS was generated through treatment with anion-exchange resin, as previously described [34,35]. The cells were stimulated with 0.1 Nm–1 µM T_3_ or 10 nM–100 µM MGL-3196 for 24 h, respectively. 

The highest concentration of DMSO was used as a solvent control. The cells were harvested, and the activities of firefly and *renilla* luciferases were determined (Dual-Glo Luciferase Assay System; Promega, Mannheim, Germany) in a Sirius luminometer (Berthold Detection Systems GmbH, Pforzheim, Germany). Firefly luciferase luminescence was normalized to *renilla* luciferase luminescence from the same transfection sample to control for differences in transfection efficiency. 

OATP1B1-mediated transport was inhibited with the competitive inhibitor glycyrrhizinic acid (GA). HEK-Co and HEK-1B1 cell lines were transiently transfected with a plasmid encoding for TRβ and cultured under TH-free conditions. The cells were treated with 100 µM GA 30 min prior to treatment with T_3_ (10 nM) or MGL-3196 (600 nM). Luciferase expression was determined after 24 h as described above.

### 4.4. Isolation and Cultivation of Primary Murine Hepatocytes

Primary murine hepatocytes (PMHs) were isolated from WT and TRβ knockout (TRβKO) mice through a 2-step perfusion procedure of the livers of 8–10-week-old male mice via the portal vein [36]. After perfusion with 37 °C prewarmed HBSS without Ca^2+^ and Mg^2+^ (Gibco™, Thermo Fisher Scientific, Schwerte Germany) containing 0.5 mM EGTA (Sigma Aldrich, Darmstadt, Germany), the livers were digested with 2 mg/mL collagenase II (Worthington Biochemical, Lakewood, NJ, USA) in a DMEM (low-glucose) (Thermo Fisher Scientific Gibco, Schwerte, Germany) supplemented with 100 U/mL penicillin, 0.1 mg/mL streptomycin, and 15 mM HEPES (Sigma Aldrich, Darmstadt, Germany). After the excision of the liver and mechanical disruption, the liver cell suspension was filtered through a 70 µm cell strainer and centrifuged for 2 min at 50× *g*. Viability was determined using a trypan blue exclusion test. Hepatocytes were seeded on collagen-coated 6-well plates at a density of 3.5 × 10^5^ cells/mL in a 1:1 mixture of William’s E Medium (Thermo Fisher Scientific Gibco, Schwerte, Germany) and low-glucose DMEM supplemented with penicillin–streptomycin, 1 µM insulin (Sigma Aldrich, Darmstadt, Germany), 1 µM dexamethasone (Sigma Aldrich, Darmstadt, Germany), and 10% of TH-depleted FCS. PMHs were treated with the respective doses of T_3_ and MGL-3196 for 24 h.

### 4.5. Gene Expression Analysis

Briefly, 8 × 10^5^ HEK-1B1 cells were seeded in a 6-well dish and grown to 60–80% confluence overnight. Transfection with the plasmids encoding for TRα or TRβ was carried out the next day as described above. After 24 h in the TH-depleted medium, HEK-1B1 cells were treated with the EC_50_ of T_3_ and MGL-3196 for another 24 h. PMHs were treated with the EC_50_ of T_3_ and MGL-3196 directly after 4 h of attachment for 24 h. The total RNA was isolated from HEK-1B1 and PMH using a QIAGEN RNeasy Mini Kit (QIAGEN, Hilden, Germany). The concentration and purity of the RNA samples were measured using a NanoDrop2000 device (Thermo Fisher Scientific, Schwerte, Germany). cDNA synthesis real-time PCR was performed on a LightCycler480 (Roche, Mannheim, Germany) [20]. For the normalization of gene expression, *18S*, *Ppia*, *β-Actin*, and *Rpl13a* were used as reference genes. Primer sequences are provided in Table 2.

### 4.6. Seahorse XF-24 Mito Stress Test

The oxygen consumption rate (OCR) was measured with an XF24 extracellular analyzer (Seahorse Bioscience, North Billerica, MA, USA). Briefly, 7.5 × 10^3^ HEK-1B1 cells were seeded in poly-l lysine-coated XF24 culture plates and cultured in a DMEM with 10% TH-depleted serum for 4 days. The cells were treated with T_3_ (100 nM) or MGL-3196 (6 µM) 48 h before OCR measurement. Reagent concentration was optimized using a Mito Stress Test kit from Seahorse Bioscience (103015-100) according to the protocol and the XF24 program’s algorithm. Oligomycin, FCCP, antimycin A, and rotenone were all used at a concentration of 1 µM. On the day of measurement, the TH-depleted medium was replaced by 500 µL of the assay medium, and the plate was incubated at 37 °C for 1 h in a non-CO_2_ incubator. 

Basal respiration was calculated as [baseline O_2_ consumption] − [OCR after rotenone and antimycin A]. ATP production corresponds to the OCR used for mitochondrial ATP synthesis via ATP synthetase, which is inhibited by oligomycin. The expression [OCR after FCCP] − [OCR after rotenone and antimycin A] determined the maximal respiration (respiratory capacity), and spare respiratory capacity was calculated as [OCR after FCCP] − [Basal OCR].

### 4.7. Statistics

We used one-way ANOVA with Tukey’s post hoc test for the statistical analysis of normally distributed datasets unless otherwise noted. Differences were considered significant when *p* < 0.05. For gene expression data, statistical significance was calculated using log-transformed data (to obtain normal distribution) as recommended by the MIQE guidelines [37]. Data were analyzed and plotted with GraphPad Prism 8 (GraphPad Software, San Diego, CA, USA).

## Figures and Tables

**Figure 1 ijms-23-13714-f001:**
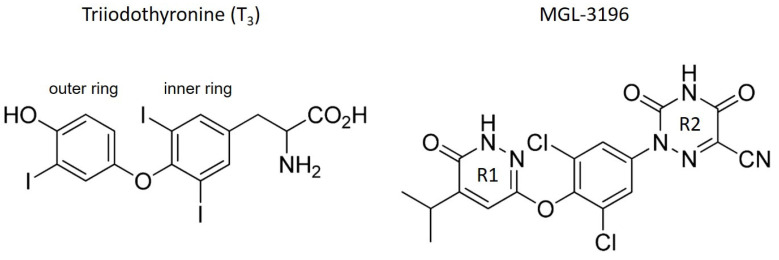
Structures of triiodothyronine (T_3_) and MGL-3196. In comparison to the structure of T_3_, the phenolic outer ring is replaced by a pyridazinone ring (R1) in MGL-3196, resulting in a heterocyclic structure. Instead of the α-amino acid group, MGL-3196 possesses an azauracil residue with a cyano group (R2) ([11]).

**Figure 2 ijms-23-13714-f002:**
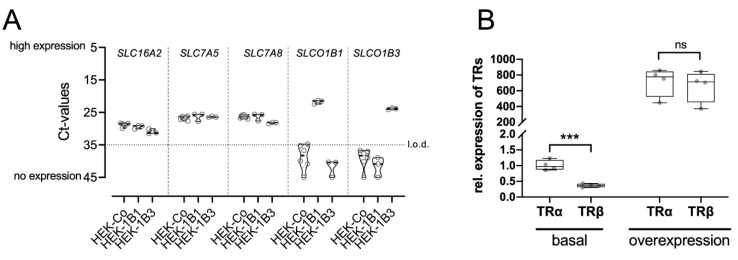
Expression of TH transporters and TRs in HEK293 cell lines: (**A**) RT-PCR analysis of genes encoding for TH transporters MCT8 (*SLC16A2*), LAT1 (*SLC7A5*), and LAT2 (*SLC7A8*) in wild-type HEK293 cells (HEK-Co) as well as in HEK-1B1 and HEK-1B3 cells stably expressing OATP1B1 (*SLCO1B1*) or OATP1B3 (*SLCO1B3*), respectively. The dotted line marks the limit of detection (l.o.d.) at Ct ≥ 35. Ct = cycle threshold; LAT = l-type amino acid transporter; MCT = monocarboxylate transporter; (**B**) expression of TRα and TRβ in HEK293 cells. Transient transfection led to similar expressions of both receptors. Student’s *t*-test: ns = not significant; *** *p* < 0.001.

**Figure 3 ijms-23-13714-f003:**
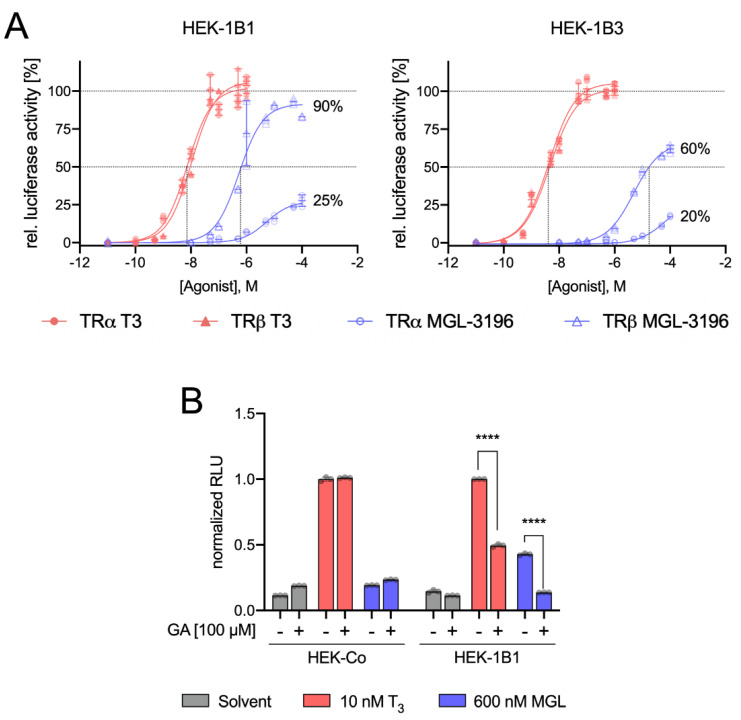
Dose-response curve and luciferase activity: (**A**) EC_50_ values were determined by DR4 luciferase reporter assay in cells stably expressing OATP1B1 or OATP1B3 (HEK-1B1 and HEK-1B3, respectively). Additionally, the cells were transfected with plasmids either encoding for TRα (circles) or TRβ (triangles). Cells were cultured in TH-depleted serum and treated with T_3_ (0.1–1000 nM; red-filled triangles or circles) or MGL-3196 (10–10,000 nM; blue open triangles or circles) for 48 h; (**B**) inhibition of OATP1B1-dependent uptake of MGL-3196 was determined in a DR4 luciferase reporter assay with HEK-Co (control) cells and HEK-1B1 cells (stably expressing OATP1B1) and 100 µM GA (glycyrrhizinic acid) as OATP1B1 inhibitor. RLU in T_3_-treated cells without GA = 1.00 in each cell line. Two-way ANOVA with Tukey’s post hoc test: **** *p* < 0.0001.

**Figure 4 ijms-23-13714-f004:**
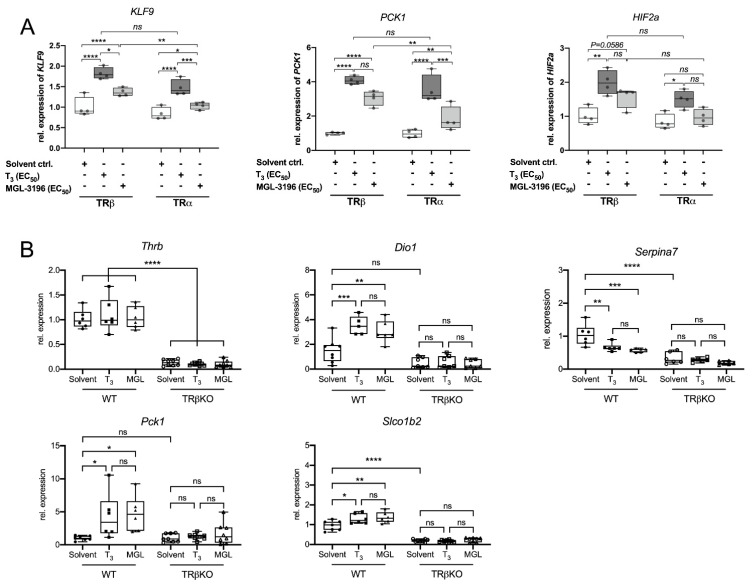
Gene expression analysis in HEK-1B1 cells and primary murine hepatocytes: (**A**) HEK-1B1 cells either expressing TRβ or TRα were treated with 10 nM T_3_ (dark grey) or 600 nM MGL-3196 (light grey) for 48 h and expression of TH target genes *KLF9* (krüppel-like factor 9), *PCK1* (pyruvate carboxylase kinase 1) and *HIF2a* (hypoxia-inducible factor 2a) was measured (n = 3); (**B**) primary murine hepatocytes of WT and TRβKO mice were treated with EC_50_ of T_3_ or MGL-3196 for 24 h and expression of hepatic TH target genes was analyzed by using RT-PCR. MGL-3196 and T_3_ induced similar changes in gene expression of *Dio1*, *Serpina7*, and *Pck1* (n = 6). Both T_3_ and MGL-3196 led to an increased expression of *Slco1b2*, the murine orthologue of the human *SLCO1B1* and *SLCO1B3*. One-way ANOVA with Tukey’s post hoc test: ns = not significant; * *p* < 0.05, ** *p* < 0.01, *** *p* < 0.001, **** *p* < 0.0001.

**Figure 5 ijms-23-13714-f005:**
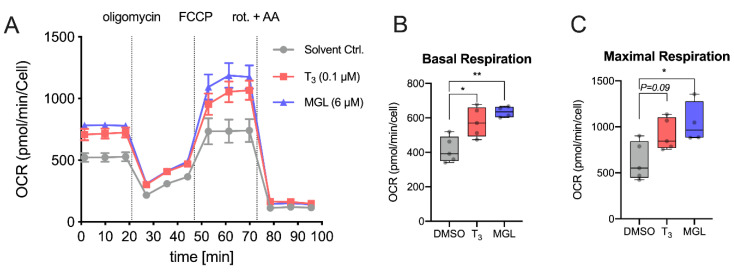
MGL-3196 increases mitochondrial activity in OATP1B1-expressing cells: (**A**) measurement of oxygen consumption rate (OCR) in HEK-1B1 cells treated for 48 h with T_3_ (100 nM) or MGL-3196 (6 µM) (n = 5 biological replicates per group); (**B**) basal OCR ([last rate measurement before oligomycin injection] − [non-mitochondrial respiration rate]); (**C**) maximal respiration ([maximum rate measured after FCCP injection] − [non-mitochondrial respiration rate]); (**D**) ATP production ([last rate measurement before oligomycin injection] − [minimum rate measured after oligomycin injection]); (**E**) proton leak ([minimum rate measured after oligomycin injection] − [non-mitochondrial respiration rate]); (**F**) spare respiratory capacity ([maximal respiration] − [basal respiration]); (**G**) non-mitochondrial oxygen consumption (minimum rate measured after rotenone/antimycin A injection). One-way ANOVA with Tukey’s post hoc test: ns = not significant; * *p* < 0.05, ** *p* < 0.01. FCCP, carbonyl cyanide-p-trifluoromethoxyphenylhydrazone; rot., rotenone; AA, antimycin A.

**Table 1 ijms-23-13714-t001:** EC_50_ values for T_3_ and MGL-3196.

	TR Isoform	Transporter	log(EC_50_) ± SD	EC_50_ [µM]
T3	TRα	OATP1B1	−8.091 ± 0.043	0.007
		OATP1B3	−8.441 ± 0.030	0.004
	TRβ	OATP1B1	−8.059 ± 0.031	0.009
		OATP1B3	−8.356 ± 0.022	0.004
MGL-3196	TRα	OATP1B1	-	-
		OATP1B3	-	-
	TRβ	OATP1B1	−6.221 ± 0.046	0.601
		OATP1B3	−4.753 ± 0.013	17.660

**Table 2 ijms-23-13714-t002:** qRT-PCR primer sequences.

	Gene	Forward Primer 5′-->3′	Reverse Primer 5′-->3′	Accession No.
*human*	*18S*	CGG CTA CCA CAT CCA AGG AA	GCT GGA ATT ACC GCG GCT	NR_145820.1
	*ACTB*	AGA GCT ACG AGC TGC CTG AC	AGC ACT GTG TTG GCG TAC AG	NM_001101.5
	*PPIA*	AAC GTG GTA TAA AAG GGG CGG	CTG CAA ACA GCT CAA AGG AGA C	NM_021130.5
	*SLCO1B1*	CAC TTG CAC TGG GTT TCC AC	AAG CCC AAG TAG ACC CTT GAA A	NM_006446.5
	*SLCO1B3*	TTT TTG GAA GGG TCT ACT TGG G	TCA TTG TCC GAT GCC TTG GTA	NM_019844.4
	*PCK-1*	GGA GAA GGA GGT GGAAGA C	GAA CAC TTG CCC TCT CTT GC	NM_002591.4
	*KLF9*	CTC CCA TCT CAA AGC CCA TTA C	TGA GCG GGA GAA CTT TTT AAG G	NM_001206.2
	*SLC16A2*	AGC TCC TTC ACC AGC TCC CTA AGC	TCC TCC ACA TAC TTC ATC AGG TG	NM_006517.5
	*SLC7A5*	GGG AAG GGT GAT GTG TCC AAT CTA	CAA GTA ATT CCA TCC TCC	NM_003486.7
	*SLC7A8*	GTG CTA TCA TCG TAG GGA AGA TC	CAC AGC CTC AGG AAC CCA G	NM_012244.4
*murine*	*18s*	CGG CTA CCA CAT CCA AGG AA	GCT GGA ATT ACC GCG GCT	NR_003278.3
	*Ppia*	CTT GGG CCG CGT CTC CTT CG	GCG TGT AAA GTC ACC ACC CTG GC	NM_008907.2
	*Actb*	GGC CCA GAG CAA GAG AGG TA	CTG GAT GGC TAC GTA CAT GGC	NM_007393.5
	*Rpl13a*	GGG CAG GTT CTG GTA TTG GA	GGG GTT GGT ATT CAT CCG CT	NM_009438.5
	*Slco1b2*	ATC GGA CCA ATC CTT GGC TTT	TTA TGC GGA CAC TTC TCA GGT	NM_020495.2
	*Pck-1*	ATC TTT GGT GGC CGT AGA CC	ATC TTG CCC TTG TGT TCT GC	NM_011044.3
	*Dio1*	GGG CAG GAT CTG CTA CAA GG	CGT GTC TAG GTG GAG TGC AA	NM_007860.4
	*Serpina7*	TGG GCA TGT GCT ATC ATC TTC A	GAG TGG CAT TTT GTT GGG GC	NM_177920.5
	*Thrb*	GGA CAA GCA CCC ATC GTG AA	ACA TGG CAG CTC ACA AAA CAT	NM_001113417.1

## Data Availability

Not applicable.

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
