# Peer review of "Cell-Specific Transport and Thyroid Hormone Receptor Isoform Selectivity Account for Hepatocyte-Targeted Thyromimetic Action of MGL-3196"

_ijms, 2022, doi:10.3390/ijms232213714_

Round 1

Reviewer 1 Report

MGL-3196 is a new TH analog looks to have a favorable overall pharmacological profile with significant lipid-lowering efficacy in the absence of adverse effects. In this study, Hönes GS et al. using HEK293 stably overexpressing human OATP1B1 (HEK-1B1) and OATP1B3 (HEK-1B3) to identified OATP1B1 as the primary transporter for MGL-3196. MGL-3196 had a higher efficacy in activating TRβ than activation of TRα. In HEK293 cells stably expressing OATP1B1 MGL-3196 had comparable effects on mitochondrial respiration as T3. Their suggest that MGL-3196’s hepatic thyromimetic action, from a combination of hepatocyte specific transport by OATP1B1 and selective activation of TRβ over TRα. Overall, this is an interesting study. But there are still several concerns need to be clarified by the authors.

1.     OATP2B1 is expressed in both liver and heart, and it is desirable to test the role of MGL-3196 on OATP2B1 simultaneously.

2.     All the bar chat figures best to replace with box-and-whiskers plots and the individual data points need added in plots, especially in figure 3A and figure 4A.

3.     In figure 5, best to label each data graph with A to G.

4.     In figure 5A, why there is significant difference between solvent and T3 & MGL-3196 treatment groups at time 0 (time 0 should means before the treatment right?)

5.     Please further clarify the how to calculate the ATP production, proton leak, spare respiratory capacity and Non-mitochondrial oxygen consumption.

6.     It would make this study more meaningful if KB-2115 (Eprotirome) or/and animal experiments were included.

Reviewer 2 Report

In this excellent work, Hones and colleagues provide new and significant insight into the mechanism of action of thyromimetic MGL-3196. They demonstrate that critical aspect of its liver specificity is based on its marked preference to be transported into cells by the OATP1B1 transporter, which is highly specific to liver tissue in humans.

Very minor comments and suggestions include:

It may be helpful in figure 3A to indicate what groups the symbols represent so that there is no need to read the legend.

Line 182: A period is missing before “Furthermore”

Line 182: What is the tissue origin of huh-7 cells?

I will be helpful to put the names of the genes on top of the graphs in figure 4A, as it is done in Fig. 4B.

Line 205: Maybe “physiological” can be substituted for “natural”.

Is OATP1B1/3 up-regulated by T3/MGL3196 in human HEK293 cells also? It could be important to determine if this positive feedback loop applies to humans also.

Round 2

Reviewer 1 Report

The manuscript is a marked improvement over the previous version in terms of writing and methodological description. The authors have responded appropriately to most of the issues raised by the reviewer.